# Does Methionine Status Influence the Outcome of Selenomethinione Supplementation? A Comparative Study of Metabolic and Selenium Levels in HepG2 Cells

**DOI:** 10.3390/nu14183705

**Published:** 2022-09-08

**Authors:** Yili Hu, Xiaocui Chai, Jun Men, Shen Rao, Xin Cong, Shuiyuan Cheng, Zhixian Qiao

**Affiliations:** 1National R&D Center for Se-Rich Agricultural Products Processing, School of Modern Industry for Selenium Science and Engineering, Wuhan Polytechnic University, Wuhan 430023, China; 2Institute of Hydrobiology, Chinese Academy of Sciences, Wuhan 430072, China; 3Research and Development Center, Enshi Se-Run Material Engineering Technology Co., Ltd., Enshi 445000, China

**Keywords:** selenomethionine, methionine restriction, metabolomics, selenium uptake, redox status

## Abstract

Methionine restriction and selenium supplementation are recommended because of their health benefits. As a major nutrient form in selenium supplementation, selenomethionine shares a similar biological process to its analog methionine. However, the outcome of selenomethionine supplementation under different methionine statuses and the interplay between these two nutrients remain unclear. Therefore, this study explored the metabolic effects and selenium utilization in HepG2 cells supplemented with selenomethionine under deprived, adequate, and abundant methionine supply conditions by using nuclear magnetic resonance-based metabolomic and molecular biological approaches. Results revealed that selenomethionine promoted the proliferation of HepG2 cells, the transcription of selenoproteins, and the production of most amino acids while decreasing the levels of creatine, aspartate, and nucleoside diphosphate sugar regardless of methionine supply. Selenomethionine substantially disturbed the tricarboxylic acid cycle and choline metabolism in cells under a methionine shortage. With increasing methionine supply, the metabolic disturbance was alleviated, except for changes in lactate, glycine, citrate, and hypoxanthine. The markable selenium accumulation and choline decrease in the cells under methionine shortage imply the potential risk of selenomethionine supplementation. This work revealed the biological effects of selenomethionine under different methionine supply conditions. This study may serve as a guide for controlling methionine and selenomethionine levels in dietary intake.

## 1. Introduction

Selenium has been considered an essential trace element for humans since the late 1950s [1]. This element exerts antioxidant, antitoxic, anticancer, and liver-protective effects at optimal doses [2,3]. However, the global distribution of selenium resources is heterogeneous [4,5]. More than 15% of the world’s population suffers from selenium deficiency and bears the risk of Kashin–Beck diseases [6]. Understanding selenium utilization and biotransformation in selenium species is important.

As a major selenium species in human dietary sources, selenomethionine (SeMet) is produced in plants and fungi and abundant in certain selenium-enriched protein products. Compared with inorganic selenium, SeMet is more recommendable in dietary selenium supplementation [7]. SeMet is not only an excellent supplier of selenium but also shows high redox activity and plays multiple roles in maintaining human health. It has broad biological effects, including antioxidant, nutraceutical, anti-inflammatory, immunity-enhancing, and cancer-preventive effects. A randomized clinical trial on 100 peripartum cardiomyopathy patients found that an oral supplement of 200 µg SeMet per day decreases the symptoms of heart failure and mortality [8]. Supplementation with SeMet can enhance the function of vessels by reducing the accumulation of inflammatory macrophages and the formation of atherosclerotic plaques [9]. SeMet also protects against viral infections by acting as an antioxidant, bolstering the host immune system, and attacking viral components [10]. The role of SeMet as a multifunctional healthcare agent is therefore worthy of investigation.

Following selenocysteine (Sec), SeMet is the second most discussed selenium species in proteins and enzymes; it is inserted into human proteins via the random substitution of methionine and gene coding [11]. Physiological consequences of SeMet uptake can be ascribed to its direct metabolization into reactive forms of selenium, such as selenide, selenoxide, and Sec, or storage as a substitute for methionine (Met) in proteins [12]. The structure of SeMet is similar to its sulfur analog Met. After being absorbed via the intestinal transport channel and delivered into cells, enzymes can hardly distinguish between Met and SeMet. Thus, SeMet often shares the same cellular pathways with Met. Nonetheless, the reaction kinetic parameters of SeMet and Met are different because the CH_3_-Se group of SeMet is more hydrophobic than the CH_3_-S group of Met [13]. Certain distinctions between their biological consequences should be emphasized.

Few studies have explored the interplay between SeMet and Met as well as their biopotencies under various relative amounts. Based on a limited number of studies, the incorporation of SeMet in JNK stimulatory phosphatase-1 in vitro would not significantly affect the expression and activity of this enzyme [14]. In green sturgeon, SeMet uptake is mediated by the same transporter(s) as Met, and the absorption kinetics are similar for both substrates [15]. SeMet can randomly replace Met in more than 85% of some human proteins, such as netrin-1, plexinB3, and EphA3 [16]. Proteins loaded with SeMet exhibit differences from their unsubstituted analogs. A previous study in rats demonstrated that the synthesis of glutathione peroxidase (GSH-Px) is highly restrained in a restricted Met (MR) diet, even with SeMet supplementation [17].

In recent decades, the MR diet has been widely recommended because of its health benefits, such as repressing cancer growth [18], improving cancer therapy [19], increasing the longevity of some species [20], and protecting against the negative effects of aging on metabolic syndrome via an FGF21 mechanism [21]. Vegan or Mediterranean diets limit Met intake. However, SeMet might be released from body stores or accumulated to a toxic level under this condition. The effects of SeMet supplementation under different Met uptake levels remain unclear to date.

With the rapid development of omics technology in the past decades, exploring the systematic biological effect of SeMet at the molecular level has become possible. In this work, the effects of SeMet supplementation on different Met supply levels were mainly obtained with the aid of NMR-based metabolomics, which is endowed with powerful reproducibility, quantitative ability, and capability for identifying compounds and deducing structures of unknowns [22]. HepG2 cell line was chosen as the model for monitoring cell metabolism due to its wide application in the evaluation of metabolic response [23,24]. Furthermore, the selenium uptake, selenoprotein gene transcription, and redox status of cells incubated with different Met and SeMet levels were explored. Therefore, this study aims to explore the biological outcomes of SeMet supplementation under different Met supply conditions to assess its safety and effectiveness. This study may serve as theoretical guidance for controlling Met and SeMet levels in dietary intake.

## 2. Materials and Methods

### 2.1. Chemicals

L-selenomethionine and L-methionine were obtained from Sigma-Aldrich (St. Louis, MO, USA). NaH_2_PO_4_·2H_2_O, K_2_HPO_4_·3H_2_O and methanol (analytical grade) were obtained from Sinopharm Chemical Reagent Co., Ltd. (Shanghai, China). D_2_O and 3-(Trimethylsilyl)propionic-2,2,3,3-d4 acid sodium salt (TSP-d4) were obtained from Cambridge Isotope Laboratories (Cambridge, MA, USA).

### 2.2. Culture and Treatment of HepG2 Cells

HepG2 cells were obtained from ATCC and cultured in regular Dulbecco’s modified Eagle’s medium (DMEM; Hyclone, Thermo, Logan, UT, USA) supplemented with 10% fetal bovine serum (FBS) (Gibco, Thermo, Logan, UT, USA). After cells in culture dishes reached 40–50% confluence, the regular medium was replaced with different DMEM culture mediums listed below, randomly.

Met-free DMEM was customized at Boster Biological Technology Co., Ltd. (Wuhan, China). All treatments were prepared by adding different amounts of Met and SeMet into Met-free DMEM with 10% FBS. The contents of both free Met and total Met in FBS were quantitated by the national standard method (GB 5009.124-2016) with an automatic amino acid analyzer (A300, MembraPure, Bodenheim, Germany). Correspondingly, SeMet or Met, 200 μM SeMet, 200 μM Met, 200 μM SeMet and 200 μM Met, 400 μM Met, and 400 μM SeMet and 200 μM Met were supplemented in the media of the MR, MR_SeMet, control (its ingredients are the same as the standard DMEM), control_SeMet, Met, and Met_SeMet groups, respectively. For cellular metabolites and total selenium analysis, cells were washed three times with phosphate buffer (pH 7.4) to remove the excess medium after being exposed for 48 h. Phosphate buffer was discarded after centrifugation, and the cell pellets were stored in a −80 °C refrigerator (DW-86L388J, Haier, Qingdao, China) before use.

### 2.3. Cell Viability Assay

After cells were seeded in 96-well plates with regular DEME medium for 12 h, the mediums were replaced with that containing different concentrations of Met and SeMet. The viability of HepG2 cells exposed to different SeMet and Met concentrations for 12, 24, and 48 h was measured using the MTT Cell Proliferation and Cytotoxicity Assay Kit (Beyotime, Beijing, China) in accordance with the manufacturer’s instructions. Each treatment was performed with three biological replicates.

### 2.4. Quantification of Total Selenium in HepG2 Cells

For the determination of total selenium, HepG2 cells were digested with concentrated HNO_3_ in a Microwave Digestion System (WX-8000, Preekem, Shanghai, China). After being diluted to 2% HNO_3_ solution, the total selenium concentrations in the cells were measured through inductively coupled plasma mass spectrometry (ICP-MS, NexION300X, PerkinElmer, Wellesley, MA, USA) using external calibration. The operational parameters of ICP-MS were as follows: nebulizer Ar gas, 0.95 L/min; auxiliary Ar gas, 1.2 L/min; plasma Ar gas, 15 L/min; RF power, 1400 W; dwell time, 50 ms; and lens voltage, 7.2 V. Each treatment was performed with three biological replicates.

### 2.5. Extraction of Intracellular Metabolites

Cell pellets were homogenized in 600 μL of methanol/H_2_O (2:1 *v*/*v*) solution and subjected to three freeze–thaw cycles. Sonication was performed at 4 °C for 15 min, with the repeat cycle of 1 min sonication and 1 min pause. The supernatant was collected after centrifugation at 12,000× *g* for 10 min, and the remaining cell residues were further extracted twice using the above procedure. The supernatant from the three extractions was pooled together and lyophilized after methanol removal using a SpeedVac (Thermo Fisher Scientific, Waltham, MA, USA). The cell extracts were dissolved in the phosphate buffer (pH 7.4, 99.9% D_2_O) containing 0.001% TSP-d4. After centrifugation at 16,000× *g* for 10 min at 4 °C, the supernatants were transferred into 5 mm NMR tubes for NMR analysis.

### 2.6. NMR Measurements of Cell Extracts

^1^H NMR spectra of cell extracts were acquired using noesygppr1d sequence on a Bruker Avance III 800 MHz NMR spectrometer equipped with a cryogenic probe (Bruker, Biospin, Germany). Water presaturation was achieved by irradiation of water resonance. All 1H NMR spectra were multiplied by an exponential function with a line broadening factor before conducting the Fourier transformation. The spectra were referenced to the methyl resonance signal (δ 0.000) of TSP. For the signal assignment, 2D NMR spectra, including ^1^H-^1^H correlation spectroscopy (COSY), total correlation spectroscopy (TOCSY) and J-resolved spectra, ^1^H-^13^C heteronuclear single quantum coherence (HSQC), and heteronuclear multiple bond correlation (HMBC), were acquired under previously described parameters and processing information [25].

### 2.7. NMR Data Processing and Multivariate Data Analysis

The spectral region δ 0.5–9.5 was integrated into regions with an equal width of 0.004 ppm using an AMIX software package (V3.9.2, Bruker Biospin, Karlsruhe, Germany). Region δ 4.606–5.166 was discarded to avoid the effect of water saturation. Each integral bucket was then normalized to the sum of the total integrals prior to statistical analysis.

Multivariate data analyses were performed on SIMCA-P+ software (V. 14.0, Umetrics, Malmö, Sweden). Principal component analysis (PCA) was first conducted on the mean-centered NMR data to generate an overview of the data distribution and detect potential outliers. Orthogonal projection to latent structure with discriminant analysis (OPLS-DA) was subsequently applied to the unit variance scaled data [26]. The quality of the model was monitored by the parameters Q2 and R2X, which indicate the predictability and the total explained variations of the model, respectively [27]. The validity of all models was further ensured by permutation tests (200 permutations) and CV-ANOVA tests (*p* < 0.05).

The model was interpreted by back-transformed loadings incorporated with color-coded correlation coefficients with Matlab script (MATLAB R2016a, The Mathworks Inc., Natwick, WY, USA). The color of the correlation coefficient indicates the contribution of the metabolite to class differentiation. In this study, a correlation coefficient |r| greater than 0.755 was considered to indicate statistical significance (f = 5, *p* < 0.05).

### 2.8. Quantitative Real-Time PCR Analysis

Cells were harvested after 48 h of incubation, and total RNA was extracted with Trizol (Transgen Biotech, Beijing, China) in accordance with the manufacturer’s protocol. Reverse transcription was performed on the ABI 7900HT system (Applied Biosystems, Foster City, CA, USA) using a First-Strand cDNA Synthesis kit (Thermo Scientific, Rockford, IL, USA). Quantitative RT-PCR was performed in triplicate using SYBR^®^ Premix Ex Taq^TM^ II (TaKaRa, Kyoto, Japan). The primers for six selenoprotein genes (*GPX1*, *GPX2*, *GPX4*, *TXNRD1*, *TXNRD2*, and *SELENOP*) and the reference gene (*GAPDH*) are listed in Appendix A. The relative mRNA expression was normalized to that of *GAPDH* and was determined using the Delta Ct method [28]. Each treatment was performed with three biological replicates and three technical replicates.

### 2.9. Cellular GSH/GSSG and NADP+/NADPH Assays

Cellular reduced glutathione (GSH) and oxidized glutathione (GSSG) levels were measured using GSH and GSSG assay kits (S0053, Beyotime Biotechnology, Shanghai, China) in accordance with the manufacturer’s instructions. Cells for kit assay were washed with PBS and collected by centrifugation. Precipitated protein removal reagent M was added to cells; samples were then subjected to two rapid freeze–thawing processes using liquid nitrogen and a 37 °C water bath. After an ice bath for 5 min, cells were centrifuged at 10,000× *g* for 10 min, and the supernatant was used for the determination of total glutathione. For the determination of GSSG, a GSH clearance reagent was added to the samples prepared above, and the vortex was conducted immediately. Then 4 µL of GSH scavenging reagent per 100 µL of the sample was added to the solution. The reaction was performed at 25 °C for 60 min. The absorbance values of this solution at 412 nm were measured by a microplate reader after adding the total glutathione assay working solution.

Oxidized nicotinamide adenine dinucleotide phosphate (NADP+) and reduced nicotinamide adenine dinucleotide phosphate (NADPH) were determined using their assay kits (S0179, Beyotime Biotechnology, Shanghai, China) in accordance with the manufacturer’s instructions. NADP+ and NADPH in cells were extracted using the extract solution in the kit. Centrifugation was performed at 12,000× *g* for 10 min at 4 °C, and the supernatant (S1) was used for supernatant (S2) preparation and total NADP+ and NADPH determination. An amount of 100 µL of the S1 was heated in a water bath at 60 °C for 30 min to decompose NADP+. After centrifugation for 5 min, the S2 in the tube was transferred for NADPH determination. An amount of 50 μL of supernatant S1 and S2 was transferred into a 96-well plate as the samples to determine total NADP+ and NADPH, and NADPH, respectively. WST-8-based colorimetric detection for NADP+ and NADPH quantitation was applied in this method [29]. Both GSH/GSSG and NADP+/NADPH assays were performed with three biological replicates and three technical replicates.

### 2.10. Statistical Analysis

Statistical analysis was performed using Graph Pad Prism 8.0 software (Graph Pad Software, LaJolla, CA, USA). Results were analyzed using Student’s t-test or one-way ANOVA, and differences between the two means were considered statistically significant at *p* < 0.05.

## 3. Results

### 3.1. Effects of Increasing SeMet and Met Supplementation on Cell Viability and Proliferation

Despite the 10% FBS contributing some free Met at a concentration of 4.70 μM in the medium, the viability of the HepG2 cells in the MR group was highly restrained. Their proliferation rate was slower than those of the cells under SeMet and Met supplementation (Figure 1). The supplementation of 200 μM SeMet under certain MR significantly improved the growth of HepG2 cells without exhibiting toxic effects on their viability. SeMet supplementation at a concentration of 200 μM under adequate and abundant Met supply conditions showed negligible impact on the viability of HepG2 cells.

Appendix A shows the viability of HepG2 cells grown on media containing various concentrations of SeMet and Met after 48 h. The proliferation of HepG2 cells increased with increasing SeMet and Met concentrations, even at 200 μM SeMet and 400 μM Met. Considering that the standard cell medium contained 200 μM Met, we used 200 μM as a supplementary unit to obtain comparable results when analyzing the effects of SeMet supplementation under restricted, adequate, and abundant Met supply conditions.

### 3.2. Effects of SeMet Supplementation on Metabolic Phenotypes of HepG2 Cells under Different Met Supply Conditions

To assess how the Met supply was associated with SeMet metabolism in HepG2 cells, we compared the metabolome of 36 samples from the six groups (MR, MR_SeMet, Control, Control_SeMet, Met, and Met_SeMet) by using an NMR-based metabolomic approach. A total of 37 intracellular metabolites were identified in two typical ^1^H NMR spectra of HepG2 extracts (Figure 2, Appendix A), the signals of which were confirmed by a set of two-dimensional NMR spectra (Appendix A). Among them, 16 were classified as amino acid-related compounds, 6 were sugar-related compounds, 3 were tricarboxylic acid (TCA) cycle-related intermediates, 3 were choline-related compounds, 3 were nucleosides, and 6 were other compounds, covering most central carbon metabolism and amino acid metabolism.

PCA was performed using the integral matrices of normalized NMR signals to generate an overview of the metabolic differences between all samples. Component 1 accounting for 62.9% of the variation and component 2 accounting for 23.5% of the variation helped further distinguish the sample groups. The score plot based on the first two principal components of the PCA model (Figure 3) showed three separate clusters of those 36 samples, in which distinctions among clusters A (MR group), B (control and control_Met groups), and C (MR_SeMet, control_SeMet, and Met_SeMet groups) were clear. Similarly, samples from different groups showed a certain trend of separation in clusters A and B, indicating distinct metabolic phenotypes generated during the 48-h incubation and exposure.

OPLS-DA models that can filter out irrelevant orthogonal signals were established to assess the metabolic responses of HepG2 cells to SeMet supplementation under restricted, adequate, and abundant Met conditions. In the OPLS-DA models, pairwise comparisons between NMR profiles of cells incubated with an additional 200 μM SeMet and those incubated with 0 μM Met (MR group), 200 μM Met (control group), and 400 μM Met (Met group) were carried out, respectively. The overall trends were similar to those observed in the PCA model but more obvious in the OPLS-DA score plots (Figure 4A,C,E). The qualities of these models were confirmed by their R2 and Q2 values and further evaluated with the *p*-value of CV-ANOVA.

The loading plots of OPLS-DA constructed between the MR and MR_SeMet groups (Figure 4B, Table 1) revealed that SeMet supplementation under MR condition significantly elevated the levels of Val, Trp, Tyr, Phe, Leu, Ile, Glu, Gln, succinate, fumarate, and acetate and decreased the levels of choline, GPC, creatine, Asp, uridine, NAD, and some sugar-related compounds, including UDP-glucose, UDP-N-acetylgalactosamine (UDP-GalNAc), and UDP-N-acetylglucosamine (UDP-GlcNAc). In cells with adequate Met supply (Figure 4D, Table 1), SeMet supplementation highly increased the levels of Val, Trp, Tyr, Phe, Leu, Ile, Glu, hypoxanthine, formate, and acetate and decreased the levels of Asp, His, UDP-glucose, UDP-GalNAc, UDP-GlcNAc, lactate, citrate, creatine, and GPC levels. In cells with abundant Met supply (Figure 4F, Table 1), SeMet supplementation led to similar consequences when compared with SeMet supplementation under adequate Met supply, except for the decrease in glycine level and less appreciable changes in His, Phe, GPC, formate, and acetate levels.

### 3.3. Selenium Uptake, Relative Transcriptional Levels of Selenoprotein Genes, and Redox Status of HepG2 Cells with SeMet Supplementation under Different Met Supply Conditions

After being exposed to SeMet under different Met supply conditions for 48 h, the total selenium content of HepG2 cells was determined using ICP-MS to further explore the uptake and metabolism efficiency of SeMet under multiple Met supply conditions. As shown in Figure 5, the average selenium content in the cells incubated with adequate Met (control group) was 5.10 pg/cell, and the restriction of Met slightly increased the uptake of selenium to an average content of 5.56 pg/cell, whereas cells with abundant Met supply had an average selenium content of 4.70 pg/cell. The total selenium content per cell significantly increased with additional SeMet supplementation, which is in line with expectations. Nonetheless, the selenium content in the cells decreased from 11,879.99 pg/cell to 3805.48 pg/cell with Met supply increasing from 0 μM to 400 μM, which indicated a declining uptake or utilization efficiency of SeMet in the culture medium.

For intracellular biological activities involved in selenium utilization, the relative transcription levels of several highly expressed selenoprotein genes in the liver cells were monitored using qRT-PCR. Figure 6 displays the relative transcription levels of *GPX1*, *GPX2*, *GPX4*, *TXNRD1*, *TXNRD2*, and *SELENOP*. Results showed that SeMet supplementation at 200 μM promoted the transcriptional activities of selenoprotein genes in the HepG2 cells. However, MR inhibited the effect of selenoprotein transcription when compared with the other groups. The transcription levels of the observed selenoprotein genes remained stable under adequate or abundant Met supply. Furthermore, SeMet supplementation with Met depletion affected the transcription of *GPX1*, *GPX2*, *TXNRD1*, and *TXNRD2* within certain limits.

GSH, GSSG, NADP+, and NADPH are important substrates that participate in biochemical reactions catalyzed by selenoenzymes of glutathione peroxidases (GPXs) and thioredoxin reductases (TRXs). GSH/GSSG and NADP+/NADPH levels were measured using commercialized kits to verify the enzymatic efficiency of GPXs and TRXs, and characterize the redox status of HepG2 cells under different SeMet and Met supply conditions. As illustrated in Figure 7A, the GSH/GSSG levels increased with the treatment SeMet, whereas no significant differences were observed among the groups of MR_SeMet, control_SeMet, and Met_SeMet. In accordance with results shown in Figure 6A–C, the transformation from GSSG to GSH was retrained in the MR group. Figure 7B shows that the NADP+/NADPH level decreased after SeMet supplementation. Meanwhile, rigorous restriction in Met supply remarkably increased NADP+/NADPH levels, which were similar to the results of *TXNRD1* and *TXNRD2* gene transcription.

## 4. Discussion

SeMet can be incorporated into proteins in place of Met [10]. In a Met-depleted environment, many cell lines, including cancer cell lines, exhibit an inability to grow, which is consistent with our results; therefore, MR could be a promising avenue to facilitate cancer treatment [30]. A strong supporting effect of SeMet under Met deficiency was confirmed in the present work. After being taken up in cells, SeMet could disguise as Met to participate in protein synthesis and follow the Met cycle and transselenation pathways to produce multiple intermediates, generating a profound impact on cell growth and biological activities [31]. Cell proliferation increased with an increasing supply of SeMet or Met, suggesting a positive dose-dependent relationship between cell growth and Met and its seleno-analog within an appropriate concentration range. In addition, the promoting effect of 200 μM SeMet verified that SeMet is a safer selenium supplement than selenite, which significantly inhibits the viability of the same cells at 3.0 μM [32].

A metabolomic approach was employed to evaluate the impact of SeMet supplementation on cell metabolic phenotypes under different Met supply conditions. Given the cluster distributions in Figure 3 and varying signals of OPLS-DA loading plots in Figure 4, their similarities and dissimilarities are worth further exploring. In different Met supply conditions, the biological consequence shared by SeMet supplementation lies in the elevation of most amino acids and the decrease in creatine, Asp, and some nucleoside diphosphate sugars, which probably resulted from either the direct reactivity of SeMet or the intracellular metabolism activity of SeMet as a pure Met analog, or both. The metabolism of SeMet is subjected to three pathways, namely, protein incorporation, transsulfuration pathway, and transamination–decarboxylation pathway [12,33]. Regardless of the selenide part, the intermediates of these pathways contain multiple metabolites that participate in central carbon and amino acid metabolism, such as alpha-ketobutyrate and alanine, which could be generated via cystathionine gamma-lyase and selenocysteine lyase, respectively [34,35]. The additional complement of alpha-ketobutyrate and alanine from the SeMet metabolism was suggested as the main contribution in the elevation of amino acids in the present study. UDP-glucuronic acid serves as a sugar donor in the glycan biosynthesis of glycoproteins and glycolipids, whereas UDP-GalNAc and UDP-GlcNAc serve as sugar donors in the O-glycan modification and O-GlcNAcylation of protein residues, respectively, which facilitate the regulation of major biological processes, including protein sorting, histone remodeling, and the subsequent cell proliferation and recognition. Mammal cells respond to excess nutrients by activating O-GlcNAcylation, which elevates the consumption of those nucleoside diphosphate sugars [36,37]. In addition, selenium supplementation correlates negatively with the activity of creatine kinase, a core enzyme in creatine metabolism, which is probably responsible for the decrease in creatine [38,39].

Compared with adequate and abundant Met supply, SeMet supplementation under Met depletion caused the most significant disturbance in metabolism pathways, including but not limited to the TCA cycle, glutamine metabolism, and choline metabolism. The amount of intermediates in the TCA cycle was higher in the cells with SeMet supplementation under Met depletion than in the other groups. The transamination–decarboxylation possibly became hyperactive without the Met buffering, and alpha-ketobutyrate possibly flowed into the TCA cycle through propanoate metabolism. Along with phosphocholine, GPC is a choline derivative and one of the two major forms of choline storage in the cytosol. A significant decline in choline and GPC levels has been suggested as a signal implying the defect of SeMet supplementation under MR because choline is essential in maintaining membrane structural integrity and promoting transmethylation metabolism and fatty acid metabolism [40]. Choline shortage can aggravate fatty liver and general liver damage [41]. Despite the HepG2 cell line not being regarded as the ideal cell model for hepatocytes [42], it shows similarity to primary human hepatocytes when levels of amino acids are relatively high [24]. High levels of amino acids were observed after 48-h SeMet exposure in our work. Therefore, the declined choline level induced by SeMet exposure might be a hint that SeMet supplementation under the MR dietary mode would incur a potential risk of fatty liver. Notably, the impact of SeMet on choline metabolism went down with increased Met supply in our study.

In the present study, the impact of SeMet on lactate, glycine, citrate, and hypoxanthine levels could be attributed to SeMet supplementation under adequate and abundant Met supply. Lactate is a hallmark of the Warburg effect; its decline is a positive signal in cancer therapy because a high level of lactate could act as an oncometabolite, immunosuppressant, and metabotoxin [43]. Lactate dehydrogenase (LDH) is a key glycolytic enzyme responsible for lactate production. A number of seleno-compounds, including ebselen and selenobenzene, are potent inhibitors of LDH [44,45]. We assumed that a similar inhibitory effect occurred after SeMet exposure with a certain amount of Met supply. Rapid glycine consumption and an active mitochondrial glycine biosynthetic pathway are strongly correlated with proliferation across cancer cells [46,47]. For the glycine decrease in cells with SeMet supplementation, the balance between glycine consumption and mitochondrial glycine biosynthetic pathway was clearly disrupted. In addition, the impact of SeMet on the TCA cycle changed in the presence or absence of adequate Met supply, confirming a noteworthy link between mitochondrial function and methionine metabolism, as revealed by Lozoya [48]. These differences in metabolic phenotypes of cells with SeMet/Met supply might help invoke the awareness of people in following the dietary intake amount of Met and SeMet.

The uptake of selenium is highly affected by cell lines, selenium species, and concentrations [32,49]. As shown in Figure 5, the uptake of SeMet was markedly affected by Met concentration. This result can be attributed to the fact that SeMet and Met share the same transport system to translocate from the culture medium into the cytoplasm [50]. With the increase in total (Se)Met content in the extracellular environment, the transport efficiency of SeMet behaved under a certain restriction. This result prompted a potential risk of selenium accumulation in the form of SeMet, which acts as a Trojan horse in a Met-deprived situation, and even a toxicological effect was not apparent within 48 h in this cellular study.

Aside from participating in cellular metabolism, selenium is also translated into selenoproteins or selenoenzymes that are catalytically active in redox processes. To clarify the potential correlation of redox status with SeMet supplementation under different Met supply, a series of selenoprotein genes were investigated because of their overwhelming roles in the oxidant defense system and selenium transportation. Enhanced selenoprotein expression could be observed in virous biological models with selenium supplementation. In the present study, the expression of *GPX1*, *GPX2*, *GPX3*, *TXNRD1*, *TXNRD2*, and *SELENOP* was upregulated with SeMet supplementation. However, the expression of these selenoproteins decreased in the MR group, and SeMet supplementation in this condition hardly elevated the expression of *GPX1*, *GPX2*, *TXNRD1*, and *TXNRD2* to the same levels as the cells supplied with adequate Met. Combined with the results of the metabolomic study, the gap in post-transcriptional activity between SeMet supplementation under MR and sufficient Met conditions was assumed to be the consequence of a certain Sec shortage because more SeMet was required in supporting cellular metabolism in the absence of Met. Hypoxanthine, which is a prevalent metabolite in the purine salvage pathway, stabilizes the RNA folding structure of riboswitches, thereby providing a mechanism for regulating gene expression in response to an increase in intracellular concentrations of the metabolite [51]. The increased hypoxanthine level in the cells with SeMet supplementation under adequate Met supply corroborated with the thriving metabolic and genetic activities.

GSH/GSSG and NADP(+)/NADPH are substrates that participate in the reactions catalyzed by selenoenzymes GPXs and TRXs, respectively. These ratios are reliable markers for evaluating biochemical parameters representative of the redox states [52]. In accordance with the cell proliferation and selenoprotein gene transcription, the values of GSH/GSSG and NADP(+)/NADPH ratios attested to an unbalanced redox status in the cells under Met starvation. This situation was greatly alleviated with SeMet supplementation irrespective of the Met supply.

## 5. Conclusions

NMR-based metabolomics, ICP-MS, and qRT-PCR were employed to unveil the metabolic phenotype, selenium uptake, and selenoprotein expressions of liver cells supplemented with SeMet under Met-deprived, adequate, and abundant conditions. SeMet supplementation elevated the levels of most amino acids and decreased the levels of creatine, Asp, and some nucleoside diphosphate sugar. SeMet supplementation under Met depletion caused remarkable selenium accumulation and the most significant disturbance in metabolites, implying a potential health risk. No toxicological effect was observed even after 48 h of treatment. Notably, the metabolic disturbance induced by SeMet declined with increasing Met supply, except for some changes in lactate, glycine, citrate, and hypoxanthine levels, which could be attributed to SeMet supplementation under adequate and abundant Met supply. The results of this work highlighted the differences in cellular response of SeMet supplementation affected by the Met status. This study could drive people’s awareness toward following the dietary intake amounts of Met and SeMet. Future studies could evaluate the toxicological effect of SeMet supplementation under MR conditions in animal models, and omics are recommended as the technique options.

## Figures and Tables

**Figure 1 nutrients-14-03705-f001:**
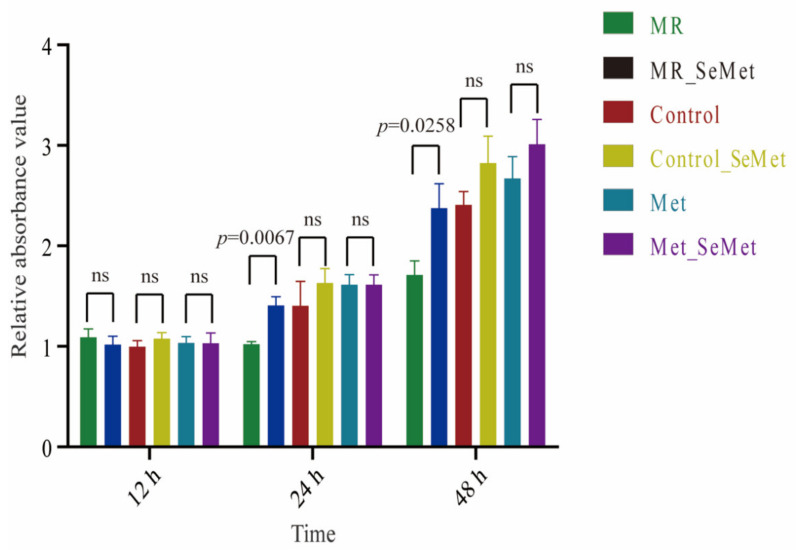
Cell viability of HepG2 cells exposed to SeMet under different Met supply conditions. Data are presented as mean ± SD; “ns” indicates not significant.

**Figure 2 nutrients-14-03705-f002:**
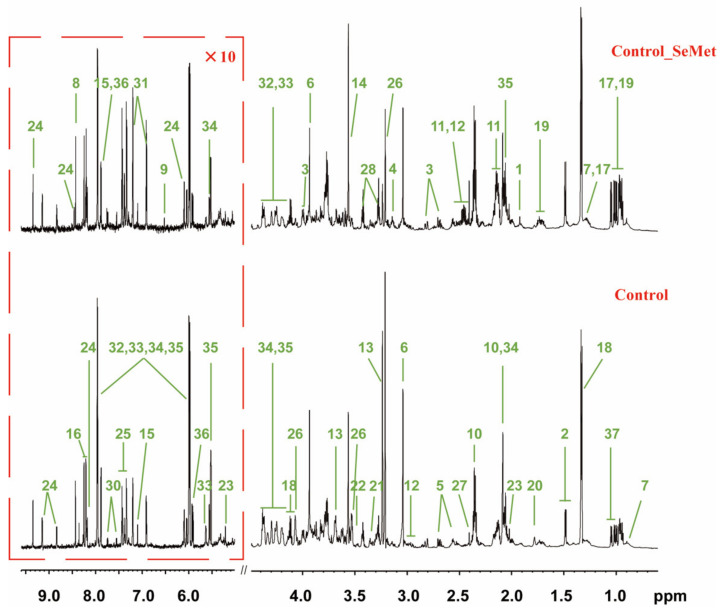
Typical 800 MHz ^1^H NMR spectra of HepG2 cell extracts from control and control_SeMet groups. The Red dashed box is vertically expanded 10 times in the spectra. Keys: 1. Acetate; 2. Alanine (Ala); 3. Aspartate (Asp); 4. Choline; 5. Citrate; 6. Creatine; 7. Fatty acid; 8. Formate; 9. Fumarate; 10. Glutamate (Glu); 11. Glutamine (Gln); 12. Glutathione; 13. Glycerophosphocholine (GPC); 14. Glycine (Gly); 15. Histidine (His); 16. Hypoxanthine; 17. Isoleucine (ILe); 18. Lactate; 19. Leucine (Leu); 20. Lysine (Lys); 21. Methanol; 22. Methylphosphate; 23. N-acetylglucosamine(GlcNAc); 24. Nicotinamide Adenine Dinucleotide (NAD); 25. Phenylalanine (Phe); 26. Phosphorylcholine (PC); 27. Succinate; 28. Taurine; 29. Threonine (Thr); 30. Tryptophan (Trp); 31. Tyrosine (Tyr); 32. UDP-glucose; 33. UDP-Glucuronate; 34. UDP-N-acetylgalactosamine; 35. UDP-N-acetylglucosamine; 36. Uridine; 37. Valine (Val).

**Figure 3 nutrients-14-03705-f003:**
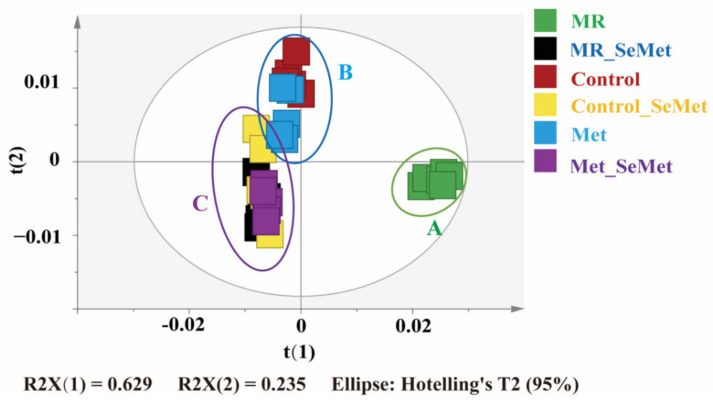
PCA scores plot derived from the ^1^H NMR data for HepG2 cell extracts from MR, MR_SeMet, control, control_SeMet, Met, and Met_SeMet groups. Samples are clustered into three separate parts (A, B, and C).

**Figure 4 nutrients-14-03705-f004:**
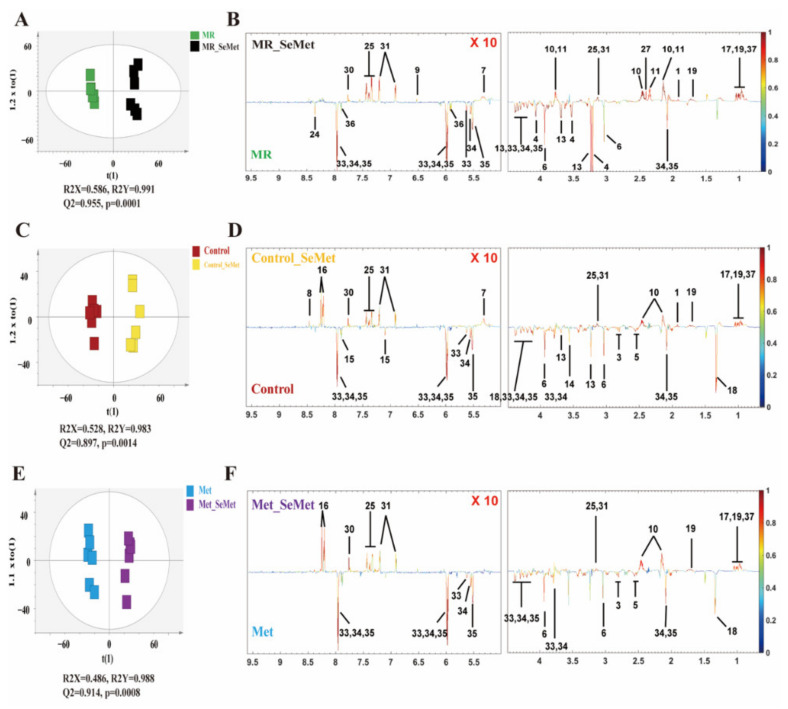
OPLS-DA scores plots (**left**) and corresponding loadings plots (**right**) derived from the ^1^H NMR data for HepG2 cell extracts: (**A**) Scores plot of the model constructed between MR and MR_SeMet groups; (**B**) Loading plot of changed metabolites between MR and MR_SeMet groups; (**C**) Scores plot of the model constructed between control and control_SeMet groups; (**D**) Loading plot of changed metabolites between control and control_SeMet groups; (**E**) Scores plot of the model constructed between Met and Met_SeMet groups; (**F**) Loading plot of changed metabolites between Met and Met_SeMet groups. Keys: 1. Acetate; 3. Aspartate (Asp); 4. Choline; 5. Citrate; 6. Creatine; 8. Formate; 9. Fumarate; 10. Glutamate (Glu); 11. Glutamine (Gln); 13. Glycerophosphocholine (GPC); 14. Glycine (Gly); 15. Histidine (His); 16. Hypoxanthine; 17. Isoleucine (ILe); 18. Lactate; 19. Leucine (Leu); 24. Nicotinamide Adenine Dinucleotide (NAD); 25. Phenylalanine (Phe); 27. Succinate; 30. Tryptophan (Trp); 31. Tyrosine (Tyr); 33. UDP-Glucuronate; 34. UDP-N-acetylgalactosamine; 35. UDP-N-acetylglucosamine; 36. Uridine; 37. Valine (Val).

**Figure 5 nutrients-14-03705-f005:**
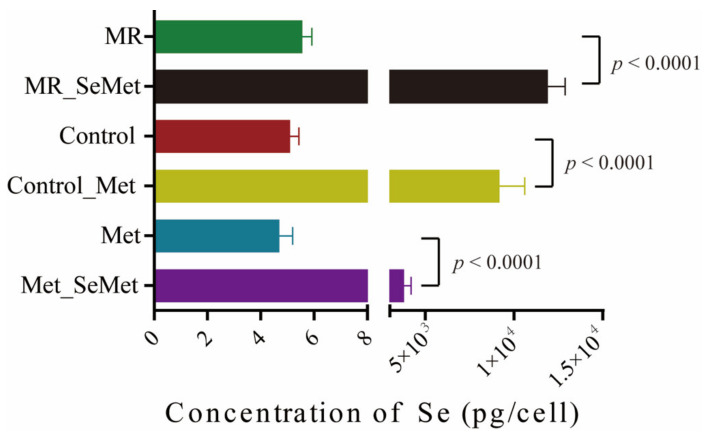
Selenium contents in HepG2 cells after 48-h exposure to SeMet under different Met supply conditions.

**Figure 6 nutrients-14-03705-f006:**
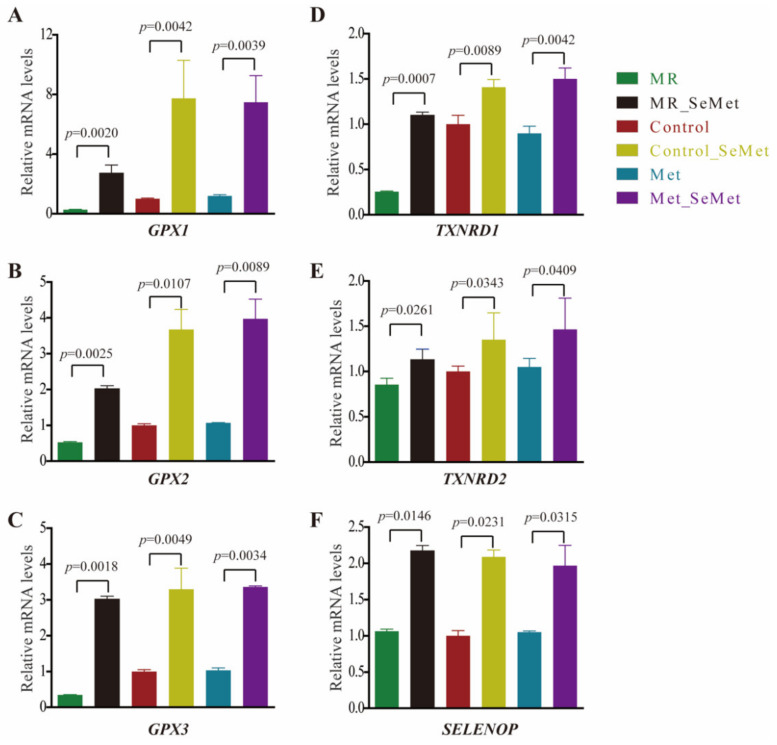
Relative transcriptional levels of selenoprotein genes in HepG2 cells after 48-h exposure of SeMet under different Met supply conditions: (**A**) Relative mRNA levels of *GPX1*; (**B**) Relative mRNA levels of *GPX2*; (**C**) Relative mRNA levels of *GPX3*; (**D**) Relative mRNA levels of *TXNRD1*; (**E**) Relative mRNA levels of *TXNRD2*; (**F**) Relative mRNA levels of *SELENOP*.

**Figure 7 nutrients-14-03705-f007:**
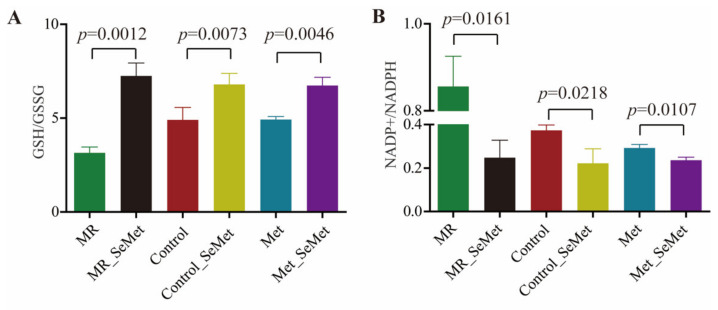
Relative contents of substrates that participate in reactions catalyzed by selenoenzymes in HepG2 cells after 48-h exposure of SeMet under different Met supply conditions: (**A**) Ratio of GSH to GSSG; (**B**) Ratio of NADP+ to NADPH.

**Table 1 nutrients-14-03705-t001:** Correlation coefficients from OPLS-DA analysis for significantly changed metabolites in corresponding cells supplemented with SeMet under different Met supply conditions.

No.	Metabolites	Correlation Coefficients		
		Model 1 ^a^	Model 2 ^b^	Model 3 ^c^
1	Acetate	**0.885**	**0.883**	−0.065
3	Aspartate (Asp)	**−0.789**	**−0.844**	**−0.933**
4	Choline	**−0.990**	−0.062	−0.051
5	Citrate	−0.491	**−0.985**	**−0.954**
6	Creatine	**−0.983**	**−0.964**	**−0.979**
8	Formate	0.708	**0.889**	0.106
9	Fumarate	**0.888**	0.345	0.650
10	Glutamate (Glu)	**0.961**	**0.922**	**0.894**
11	Glutamine (Gln)	**0.890**	0.632	0.389
13	Glycerophosphocholine (GPC)	**−0.980**	**−0.814**	−0.629
14	Glycine (Gly)	0.583	−0.737	**−0.825**
15	Histidine (His)	−0.421	**−0.829**	−0.736
16	Hypoxanthine	0.045	**0.829**	**0.918**
17	Isoleucine (Ile)	**0.964**	**0.928**	**0.843**
18	Lactate	−0.717	**−0.901**	**−0.875**
19	Leucine (Leu)	**0.979**	**0.904**	**0.823**
24	NAD	**−0.801**	−0.342	−0.287
25	Phenylalanine (Phe)	**0.983**	**0.849**	0.737
27	Succinate	**0.944**	0.405	0.162
30	Tryptophan (Trp)	**0.844**	**0.922**	**0.975**
31	Tyrosine (Tyr)	**0.977**	**0.879**	**0.777**
33	UDP-Glucuronate	**−0.964**	**−0.950**	**−0.948**
34	UDP-N-acetylgalactosamine	**−0.976**	**−0.969**	**−0.950**
35	UDP-N-acetylglucosamine	**−0.989**	**−0.944**	**−0.951**
36	Uridine	**−0.814**	−0.176	−0.062
37	Valine (Val)	**0.956**	**0.925**	**0.849**

^a^ OPLS-DA model of SeMet supplementation under MR condition (Model 1); ^b^ OPLS-DA model of SeMet supplementation under adequate Met supply condition (Model 2); ^c^ OPLS-DA model of SeMet supplementation under abundant Met supply condition (Model 3); correlation coefficient values highlighted in bold (|r| ≥ 0.755) are significant in the model.

## Data Availability

Not applicable.

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
