# Peer review of "Does Methionine Status Influence the Outcome of Selenomethinione Supplementation? A Comparative Study of Metabolic and Selenium Levels in HepG2 Cells"

_nutrients, 2022, doi:10.3390/nu14183705_

Round 1

Reviewer 1 Report

In this manuscript, Hu et al. analyzed the influence of methionine and selenomethionine concentrations on the metabolome of the liver cell line HepG2 via NMR metabolomics, transcriptomics of selected genes, and glutathione and NADP+ content measurement.

This work is interesting but too preliminary for publication as such. The authors measured the effect of methionine and selenomethionine supplementation on HepG2 cells at very short time points 24-48 h while a much longer (several cultural cycles) treatment period would bring more information that could support the conclusions (effect of selenomethionine supplementation in humans) drawn in this current manuscript. This point is raised by the authors themselves at the end of the conclusion. Therefore I cannot recommend publication in nutrients

Additional comments:

  • The authors treated cells in the presence of 10% FBS, what is the concentration of methionine in this 10% FBS media? The authors could consider using dialyzed FBS to have a more accurate/reliable concentration of methionine in the experiments after supplementation.
  • Legend of Figures 1, 5, and 6, HepG2 is written as hepG2.
  • All the NMR spectra should be provided.
  • In Figure 3, the authors should consider changing the colors of the “MR_SeMet” and “Met_SeMet” conditions are they are indistinguishable.
  • Lines 244-245, group B should be control and Met according to Figure 3.
  • Why was the NMR analysis done after 24 h treatment but all subsequent experiments after 48 h treatment?
  • The authors should indicate the exact p-values in the figures or figure legends.

Reviewer 2 Report

The manuscript by Hu et al. entitled “Biological effects of selenomethionine…” reported on the biological effects of selenomethionine under varying methionine concentrations as deprived, adequate, and abundant, which may help control methionine and selenomethionine levels in dietary intake.

There are several questions largely involving clarity.

Lines 24-26. What does “aggravated markedly” mean?

Line 58. What does “merely distinguish” mean?

Lines101-102. What does the statement that “the medium was randomly replaced” mean?

Line 112. After being seeded for 12 hours, cells in 96-well plate were replaced with media. Does this mean conditioned media was replaced with fresh media or that cells were replaced?

Line 131. What does “sonication in snatches was performed” mean?

Figures and tables require better resolution.

Line 295. Numbers could be simplified to ng/cell.

Line 449. What does “…in several cultural cycles and models in vivo” mean?

Reviewer 3 Report

The manuscript has irregularities and inaccuracies in the text. The authors need to improve the description of methods and improve the quality of discussion.

The Authors need to explain why they used HepG2. As a cellular model of what ?

HepG2 cell line was filed in1980 by researchers at the Wistar Institute. According to molecular profiling data, the HepG2 cell line can serve as a model for hepatoblastoma. The HepG2 cells show differences between normal hepatocytes and tumor cells, for which significant changes in the epigenetic regulation of nuclear and mitochondrial genes are observed [doi.10.3390/ijms222313135]

Line 446 -  Authors reported   “Results of this work highlighted the different biological effects of SeMet supplementation on liver cells under various Met supply conditions” but this sentence in not correct.

The work was to be done on normal hepatocytes. In HepG2, as hepatoblastoma model cell line, the data obtained have poor scientific value.

The title should be appropriately modified.

 Line 169 “2.8. Quantitative Real-time PCR Analysis” , I suggest putting primers as a table

 Line 188   “2.9. Cellular GSH/GSSG and NADP+/NADPH Assays” , must be indicated the type of assay carried out in detail is not enough "assay kit"

 Figure 1 is not clear must be explained and redone clearly. All figures need to be more understandable and of better quality.

Round 2

Reviewer 1 Report

The authors have addressed my concerns in this revised version. This manuscript can now be published in nutrients.

Author Response

Thank you for your recommendation.

Best Regards.

Reviewer 3 Report

The Manuscript has been revised by Authors and this new version has been improved.

Author Response

Thank you for your approval.

Best Regards.